# Enzymatic Dissociation Induces Transcriptional and Proteotype Bias in Brain Cell Populations

**DOI:** 10.3390/ijms21217944

**Published:** 2020-10-26

**Authors:** Daniele Mattei, Andranik Ivanov, Marc van Oostrum, Stanislav Pantelyushin, Juliet Richetto, Flavia Mueller, Michal Beffinger, Linda Schellhammer, Johannes vom Berg, Bernd Wollscheid, Dieter Beule, Rosa Chiara Paolicelli, Urs Meyer

**Affiliations:** 1Institute of Pharmacology and Toxicology, University of Zurich-Vetsuisse, CH-8057 Zurich, Switzerland; juliet.richetto@uzh.ch (J.R.); flavia.mueller@uzh.ch (F.M.); Urs.meyer3@uzh.ch (U.M.); 2Core Unit Bioinformatics, Berlin Institute of Health, Charité-Universitaetsmedizin, 10117 Berlin, Germany; dieter.beule@bihealth.de; 3Institute of Molecular Systems Biology and Department for Health Sciences and Technology, ETH Zürich, 8092 Zurich, Switzerland; marc.van-oostrum@brain.mpg.de (M.v.O.); bernd.wollscheid@imsb.biol.ethz.ch (B.W.); 4Institute of Laboratory Animal Science, University of Zurich, 8952 Schlieren, Switzerland; stanislav.pantelyushin@uzh.ch (S.P.); michalmateusz.beffinger@uzh.ch (M.B.); linda.schellhammer@uzh.ch (L.S.); johannes.vomberg@uzh.ch (J.v.B.); 5Neuroscience Centre Zurich, University of Zurich and ETH Zurich, 8092 Zurich, Switzerland; 6Max Delbrück Center for Molecular Medicine in the Helmholtz Association (MDC), 13125 Berlin, Germany; 7Department of Biomedical Sciences, University of Lausanne, Rue du Bugnon 7, 1005 Lausanne, Switzerland; rosachiara.paolicelli@unil.ch

**Keywords:** microglia, astrocytes, neurons, enzymatic digestion, single-cell sequencing, protocol, microglia isolation

## Abstract

Different cell isolation techniques exist for transcriptomic and proteotype profiling of brain cells. Here, we provide a systematic investigation of the influence of different cell isolation protocols on transcriptional and proteotype profiles in mouse brain tissue by taking into account single-cell transcriptomics of brain cells, proteotypes of microglia and astrocytes, and flow cytometric analysis of microglia. We show that standard enzymatic digestion of brain tissue at 37 °C induces profound and consistent alterations in the transcriptome and proteotype of neuronal and glial cells, as compared to an optimized mechanical dissociation protocol at 4 °C. These findings emphasize the risk of introducing technical biases and biological artifacts when implementing enzymatic digestion-based isolation methods for brain cell analyses.

## 1. Introduction

The neuroscience field is continuously implementing novel techniques for gene sequencing, epigenetic analyses, and proteotype profiling at a rapidly increasing rate. Obtaining single-cell suspensions from intact brain tissue is essential for downstream cell-specific transcriptomic and proteotype analyses. The main techniques available to obtain single-cell suspensions from brain tissue are enzymatic digestion (ED) and mechanical dissociation (MD). The latter is typically carried out at 4 °C [1], whereas ED requires incubation at temperatures between 30–37 °C [2,3,4]. During the dissociation process, the cells are exposed to and confronted with non-physiological conditions, which in turn can lead to cell death and/or changes in cell morphology. Although the loss of cells and changes in cell morphology are likely to occur in both ED and MD, we hypothesize that at warm temperatures, as opposed to 4 °C, surviving cells may overreact to the non-physiological milieu containing shredded extracellular matrix, cell debris, and cytoplasmic leakage [5]. Furthermore, the ED technique is associated with marked temperature shifts occurring between the eventual cell isolation step and the preceding perfusion, the latter of which is typically carried out at cold temperatures to remove blood contamination from the brain. Thermal shocks resulting from temperature shifts have been shown to elicit widespread transcriptional changes, even within short time windows [6]. Hence, unlike MD at cold temperatures, which largely spares the cellular transcriptome and proteome [7], the ED technique may introduce additional biological biases to the transcriptional and proteotype profiles of cells. Furthermore, the latter technique might also influence the relative amount of cell surface markers, as changes in surface receptor trafficking might occur during enzymatic digestions at a temperature where cells are metabolically active. This, in turn, may introduce additional bias in cell sorting analyses such as fluorescence-activated cell sorting (FACS) and cytometry by time of flight (CyTOF).

A few studies have already pointed to the possibility that ED can lead to substantial artifacts in transcriptomic analyses [8,9,10], while it remains largely unknown whether ED introduces a similar bias in proteotype profiling (e.g., in flow cytometry and mass-spectrometry analyses). Nevertheless, this issue has only been considered for transcriptional analyses, whereby possible biological consequences of ED on the proteotype profiling (e.g., flow cytometry and mass-spectrometry analyses) have up to today not been taken into consideration. Importantly, the use of transcription/translation inhibitors does not prevent the biological downregulation of RNAs and proteins that might occur during ED, nor can they prevent unwanted alterations in epigenetic marks or internalization/externalization of receptors which might be crucial readouts for certain cell-specific analyses. Given that at present, ED is the most widely used method to obtain single-cell suspensions from brain tissue (Appendix A), it is pivotal to understand the extent of cell responses when exposed to ED as compared to a method that holds cells in a quiescent state.

In the present study, we systematically investigated the influence of different cell isolation protocols on transcriptional and proteotype profiles in mouse brain tissue, thereby taking into account single-cell transcriptomics, proteotypes of microglia and astrocytes, and flow cytometric analysis of microglia. By comparing the transcriptional and proteotype profiles of brain cells isolated via ED at 37 °C and via MD at 4 °C, our study provides a comprehensive transcriptomic and proteotype resource that can be used to identify and correct for artifacts in previously published data sets generated via ED. In the present work, we also compared the two protocols in terms of their efficiency in producing single-cell suspension of healthy mouse brain tissue and mouse glioblastoma tissue. The latter was included to extend our analyses to neuropathological brain specimens containing necrotizing tissue that is surrounded by anaplastic cells. Taken together, we hereby offer an optimized and easy-to-establish MD protocol at 4 °C, which is cost-effective and technically easier to implement than conventional ED-based protocols.

## 2. Results

### 2.1. Enzymatic Tissue Digestion Induces Transcriptional Biases in Brain Cells

The large majority of MD procedures are performed at 4 °C, whereas ED is typically conducted at 37 °C. Hence, contrary to ED, MD does not require 37 °C. Therefore, we chose to compare MD at 4 °C (MD4°) with ED at 37 °C (ED37°) to take into account the most frequently used protocols for obtaining single-cell suspensions, thereby considering the dissociation/digestion procedures and their temperature components as a characterizing entity of either protocol.

First, we compared single-cell suspensions obtained by 30-min ED at 37 °C (ED37°) or MD at 4 °C (MD4°) from mouse hippocampal tissue (Figure 1a; the MD4°-protocol was optimized in-house, see Material and Methods and Appendix A for the detailed protocol). The hippocampus was selected because it represents a discrete brain region that can be dissected consistently. Visual inspection of the isolated cells showed that ED37° considerably affected cell morphology as compared to MD4°, with ED37°-processed cells being consistently smaller in size (Figure 1a).

We then conducted single-cell RNA-sequencing (scRNA-seq) to compare the transcriptional profile of hippocampal cell suspensions generated via either technique. The quality check for the scRNA-seq is provided in Appendix A. Clustering using specific gene-set enrichment for cell-type identification [2,11] (Figure 1b) revealed quantitative and qualitative differences in cellular subpopulations between ED37° and MD4°. Compared with MD4°, tissue processing via ED37° led to an increasing degree of cell death in specific cell populations, including neurons and astrocytes, shifting the balance towards a higher proportion of microglial cells in the ED37° (Figure 2b). Indeed, the cell population ratios seen in the MD4° condition were more reminiscent of the known cell densities existing in a typical murine brain as compared to the ED37° condition [12]. The differential gene expression analysis showed that ED37° caused significant transcriptional changes in microglia (226 deregulated genes), astrocytes (290 deregulated genes), and neurons (771 deregulated genes), (Figure 2a). We also found transcriptional deregulation in other brain cell types, including oligodendrocytes (369 genes), endothelial cells (128 genes), border associated macrophages (134 genes), neuronal precursor cells (121 genes), fibroblast-like cells (480 genes), and mural cells (223 genes) (Appendix A; a complete list of differentially expressed genes is provided in Appendix A). Gene ontology (GO) analysis revealed that ED37° induced global deregulation in genes associated with e.g., RNA-editing, translation, metabolic functions in most cells, and also immune pathways deregulation in microglia cells (Figure 2c, Appendix A contains the complete GO-analysis). In addition, ED37° led to the emergence of a distinct microglial subpopulation that was barely represented in cell suspensions obtained via MD4° (Figure 2b). The latter subpopulation was characterized by increased expression of immediate early genes such as *Jun* and *Fos* (Figure 2b), as well as *Egr1*, *Hspa8*, and *Jund* (Appendix A), reflecting an immediate microglial response to the ED37°. Increased immediate early gene expression, along with a deregulated expression of genes encoding for ribosomal and mitochondrial proteins (*Rpl*, *Rps*, and mt-genes), was a common feature for most cell populations exposed to the ED37° condition (Appendix A). Deregulated expression of genes encoding for ribosomal and mitochondrial proteins has also been observed in peripheral tissues following ED37° [13], akin to what we report here for brain tissue.

### 2.2. Enzymatic Tissue Digestion Causes Widespread Proteotype Artifacts in Microglia and Astrocytes

Despite the increasing number of studies performing proteotype analysis of brain cells freshly isolated via ED37° [14,15,16], it remains unexplored whether ED37° introduces biological artifacts in proteotype profiles of brain cells. Therefore, we investigated whether the type of cell isolation method could also alter the proteotype of brain cells. To this end, we focused on astrocytes and microglia (Figure 3a), as the majority of existing proteotype studies have concentrated on these cell types [17]. Notably, proteotype analysis of freshly isolated brain cells is a current challenge in the field [17]. We here demonstrate that our MD4° protocol, followed by the S-trap method for protein extraction and peptide preparation, enables solid mass spectrometry analyses of microglia and astrocytes extracted from small brain specimens such as adult mouse hippocampi (see Material and Methods).

Consistent with the effects on the transcriptome (Figure 2), proteotype analysis using data-independent acquisition (DIA)-based liquid chromatography–tandem mass spectrometry (LC–MS/MS) revealed marked differences between the proteotype of microglia (Figure 3b) and astrocytes (Figure 3c) isolated via ED37° or MD4°. In microglia, 1619 proteins were significantly different between ED37° and MD4°. For astrocytes, we found 1984 proteins to be significantly altered following ED37° as compared to MD4°. GO analysis of deregulated microglial proteins revealed that ED37° changed proteins involved in cell motility, endocytosis, and immune processes, as well as proteins pertaining to mRNA editing, histone modifications, and chromatin architecture (Figure 3b). In astrocytes, ED37° induced alterations in proteins associated with various metabolic processes, and with modifications in the translational and transcriptional machinery, similar to the effects on the microglial proteotype (Figure 3c).

We identified a remarkable consistency and correspondence between the effects of ED37° on transcriptomic and proteotype changes in both glial cell types. In fact, the top 50 deregulated RNAs significantly correlated with changes of the corresponding proteins in microglia (Figure 3d, R = 0.49, *p* = 0.00027) and astrocytes (Figure 3e, R = 0.29, *p* = 0.043). We further examined whether perfusion at RT and subsequent ED at 37 °C induced the same degree of proteotype alterations in microglial cells as compared to perfusion with cold buffers and subsequent ED37° or MD4° (Appendix A). These analyses confirmed that the proteotype changes induced by ED37° were independent of the range of thermal shock between perfusion temperature and subsequent dissociation step at 37 °C (Appendix A). Taken together, our findings thus show that ED37° induces cell responses that lead to a substantial alteration in glial cell proteotype. A complete list of deregulated proteins and GO analyses are provided in Appendix A.

### 2.3. Enzymatic Tissue Digestion Alters the Detection of Classical Microglial Markers by Flow Cytometry

Next, we examined the influence of different cell isolation techniques by flow cytometry (FC). While ED is frequently used to produce single-cell suspensions for subsequent flow cytometric analysis of microglia [18,19,20], no study has yet examined how cell responses during ED37° might influence subsequent FC outcomes. Therefore, we compared whether ED37°, relative to MD4°, alters the expression of classical immune markers used in FC-based analyses of microglia. Flow cytometric experiments confirmed that our MD4° protocol yielded cells of larger size than ED37°, as demonstrated by a significantly higher forward scatter (Figure 4a–c; gating strategy in Appendix A). We then compared the relative expression levels of the commonly used microglial markers CD45, CD11b, SIRPα, and FcγR1 between the two methods, thereby taking into account surface and intracellular marker expression. Surface expression was significantly increased for CD11b, CD45, and SIRPα in ED37°-isolated cells compared to cells obtained via MD4° (Figure 4d–g). The increase in the intracellular staining of CD11b indicates a significant internalization after ED37° (Figure 4h,i), consistent with the increase in endocytosis-related proteins observed in the proteotype analysis (Figure 3). Moreover, intracellular CD11b expression was the main discriminating factor when the MD4° and ED37° conditions were clustered together (Figure 4h). These findings thus show that the cell isolation method can influence the cellular indices used to select and study microglial cell populations in FC analysis. Of note, the present study demonstrated that our MD4° protocol yielded a higher percentage of microglial singlets and live cells as compared ED37° (Appendix A), showing that enzymes are not strictly necessary to obtain a single-cell suspension.

Finally, we also compared the two techniques for their efficiency in producing live single-cell suspensions of CD45^low^/CD11b^+^ cell populations from murine GL-261 glioma tumor tissue, which can be more challenging to dissociate than healthy brain tissue. We found that the two techniques gave comparable results in terms of both percentages of CD45^low^/CD11b^+^ singlets and live cells from tumor-bearing brain hemispheres (Appendix A). Akin to the analyses conducted in healthy mouse brain tissue, these results demonstrate that ED is not a prerequisite for efficient dissociation of brain tumor tissue.

## 3. Discussion

Our study aimed at systematically investigating the possible influence of different cell isolation protocols on transcriptional and proteotype profiles in mouse brain tissue, thereby taking into account single-cell transcriptomics of brain cells, proteotypes of microglia and astrocytes, and flow cytometric analysis of microglia. Our findings indicate that ED37°-based cell isolation induces alterations in gene and protein expression involving both up- and downregulations. We found that 30 min ED37° was associated with altered expression of genes related to cellular metabolic processes and energy expenditure pathways in both microglia and astrocytes. Moreover, pathways associated with the immune system were also altered following ED37°. In particular, in microglia cells, transcripts associated with innate immune response, major histocompatibility complex II (MHCII) protein complex binding, monocyte chemotaxis, and chemokine binding were deregulated (Appendix A). This is indicative of microglial immune activation upon ED37 °C, possibly arising in response to exposure to the non-physiological microenvironment caused by the ED-based protocol. Astrocytes isolated via ED37° were characterized by transcriptional deregulations associated with metabolic pathways, especially lipid metabolism, glutathione metabolism, and mitochondrial respiration (Appendix A). Alterations in immune-related pathways were also found in astrocytes following ED37°, which included altered cytokine receptor activity, MyD88-dependent toll-like receptor signaling pathway, and MHC protein binding (Appendix A). The proteotype analysis of microglia and astrocytes showed that many of these transcriptional changes are mirrored at the translational level (Figure 3d,e). In both microglia and astrocytes, ED37° induced marked deregulation in proteins associated with transcription and translation (Appendix A). Moreover, microglia isolated via ED37° responded with changes in proteins associated with cell migration and motility, immune system process, leukocyte cell-cell adhesion, and response to chemical stimulus (Appendix A). This points towards ED37°-induced microglial activation at the translational level, akin to what we observed at the transcriptomic level. Finally, the proteotype analysis revealed deregulations in proteins associated with epigenetic modification such as chromatin organization and histone modifications (e.g., histone acetylation and methylation, Appendix A).

We hypothesize that the ED37°-induced changes may be caused by the thermal shock suffered by the cells and may mirror their response to the non-physiological microenvironment present during ED. It is important to point out that with the present dataset, it is not possible to dissect which of the cellular responses are due to the thermal shock and which are caused by the enzymatic digestion and its microenvironment per se. However, these alterations are likely to introduce undesirable biological biases in cell-specific transcriptomic and proteotype profiling of brain specimens. While the potential risk of introducing biological artifacts through the use of ED37°-based cell isolation techniques has received recent attention in the context of transcriptomic studies [8,9,10], our work is the first to systematically and specifically address this issue and to extend this concern to flow cytometry and proteotype analyses. The latter is of particular importance given the increasing use of ED37° in mass-cytometry to study and characterize brain immune cells [20,21,22].

Efforts towards minimizing bias in transcriptomic datasets are indispensable not only for enhancing the reproducibility of findings, but also to avoid biological misinterpretation of datasets within studies. One possible way to prevent such biases is the use of single nucleus sequencing (snSeq), where whole tissue is homogenized and cells are lysed to release single nuclei that can be processed for sequencing [23,24]. This method is increasingly used for frozen human brain tissue samples [25] and represents a valuable alternative for single-cell characterization of brain cells, albeit to a lower sequencing depth as compared to ordinary scSeq [23]. The use of transcription and translation blockers is another possible way to reduce transcriptomic and prototype bias in ED-based protocols [9,10]. However, while this strategy might be efficient in preventing de novo gene and protein expression, it does not prevent degradation/downregulation of pre-existing RNAs and proteins, which according to the present data, readily occurs during ED37°-based isolation. Moreover, our proteotype analysis of microglia and astrocytes revealed that ED37° induced the deregulation of proteins associated with various epigenetic processes, including chromatin remodeling, histone modifications, and DNA methylation (Figure 3, Appendix A). ED37° thus likely modifies the epigenetic machinery, which cannot be prevented via transcription and/or translation blockers. The latter also falls short in not preventing changes in flow cytometric analyses of brain cells such as microglia. Indeed, here we show that ED37° leads to an increase in proteins associated with endocytosis (see GO table in Figure 3b) and to a marked internalization of a commonly used microglial cell marker (CD11b) as compared to MD4°.

While both MD and ED may be associated with a certain degree of technical bias, it is likely that keeping the cells at 4 °C (as pertaining to the MD technique) better preservers their transcriptional and proteomic state due to cellular metabolic inactivation. Indeed, both MD and ED readily alter the surface of cells due to mechanical stress and enzymatic trimming respectively, which per se introduces a degree of technical bias in both cases. However, holding cells in a metabolically quiescent state will prevent processes such as endocytosis and degradation that might further alter the intracellular/surface protein levels. Importantly, our study also provides evidence that the MD at 4 °C is efficient for obtaining single-cell suspensions from mouse brain tumor tissue (Appendix A), which is often considered to be more challenging than a healthy mouse brain specimen in terms of tissue dissociation and cell isolation.

Our findings corroborate the data generated by van den Brink et al. [13], who identified transcriptomic bias in peripheral tissue cells isolated via ED37° (e.g., pancreatic cells and muscle stem cells). This consistency is particularly remarkable when considering the ED37°-mediated deregulation in genes associated with translation, RNA processing, and cellular metabolism. The study by van den Brink et al. [13] has been used by the authors of the Tabula Muris as a reference database to remove ED37°-induced genes a posteriori from the single-cell transcriptional atlas produced in various mouse organs [26]. However, the work by van den Brink et al. [13] only considered genes that were upregulated upon ED37°, while not accounting for a potential downregulation of genes. Our study shows that the latter can occur as well. For instance, 586 out of the 771 significantly deregulated genes were found to be downregulated in neurons, and 118 out of the 226 significantly deregulated microglial genes were downregulated as well (Appendix A). Likewise, according to our proteotype analysis, ED37° caused downregulation of 772 proteins in microglia and of 1230 proteins in astrocytes (Appendix A). Moreover, in addition to the general consistency between our findings and those reported by van den Brink et al. [13], we identified several brain cell-specific alterations in biological processes after ED37°, which could not have been predicted by analyses of peripheral cells. A recent study by Ayata et al. also raised this issue by comparing microglia isolated via ED with microglial specific mRNA isolated via the translating ribosome affinity purification (TRAP) method [8]. Consistent with our findings, Ayata et al. [8] noted increased expression of immediate early genes and immune-related genes in their bulk sequencing of freshly isolated microglial cells. It thus appears that ED37°-induced artifacts not only emerge in scSeq, but also bulk sequencing.

In summary, although any brain cell isolation method available at present is likely associated with a certain degree of technical and biological artifacts, MD4° of brain tissue may readily help minimize the profound alterations in the transcriptome and proteotype of specific brain cells, as seen after standard ED at 37 °C. Our dataset provides a mean of identifying brain cell-specific genes and proteins affected by ED37°, which in turn may serve to correct bias in previously published data sets produced via ED37°. Finally, we hereby offer a cost-effective, easily established, and optimized MD protocol at 4 °C which may be useful for the neuroscientific community.

## 4. Material and Methods

### 4.1. Animals

Nine to ten weeks old male C57BL6/N mice (Charles River Laboratories, Sulzfeld, Germany) were used throughout the study. They were caged 3–5 animals per cage in individually ventilated cages (IVCs). The animal vivarium was a specific-pathogen-free (SPF) holding room, which was temperature- and humidity-controlled (21 ± 3 °C, 50 ± 10%) and kept under a reversed light–dark cycle (lights off: 09:00 AM–09:00 PM). All animals had ad libitum access to the same food (Kliba 3436, Kaiseraugst, Switzerland) and water throughout the entire study. All procedures described in the present study were approved by the Cantonal Veterinarian’s Office of Zurich under the license 063/18 (approved 01/05/2018).

For the glioma model, six to eight weeks old C57BL/6JRj male mice (Janvier Labs, Le Genest-Saint-Isle, France) were used for the inoculation of the GL-261 brain tumor cell line. They were caged 3–5 animals per cage in individually ventilated cages (IVCs). The animal vivarium was an SPF holding room, which was temperature- and humidity-controlled (21 ± 3 °C, 50 ± 10%). All animals had ad libitum access to the same food and water throughout the entire study. All procedures described for the glioma model were performed according to institutional guidelines and approved by the Swiss Cantonal veterinary office (licenses ZH246/15, approved 27 June 2016).

### 4.2. GL-261luc Cells

The murine GL-261 brain tumor cell line (syngenic to C57BL/6) was stably transfected with pGl3-ctrl and pGK- Puro (Promega, Dübendorf, Switzerland) and selected with puromycin (Sigma-Aldrich) to generate luciferase-stable GL-261 cells. Cells were cultured at 37 °C 5% CO_2_ in DMEM supplemented with 10% heat-inactivated fetal calf serum, 1% L-glutamine (Thermo Fisher Scientific, Waltham, MA, USA), and 1% Pen-Strep (Sigma, St. Luois, MO, USA).

### 4.3. Surgical Procedures

For glioma inoculation, 6–10 weeks old mice were anesthetized using a mixture of fentanyl (Helvepharm AG, Thurgau, Switzerland), midazolam (Roche Pharma AG, Basel, Switzerland), and medetomidine (Orion Pharma AG, Zug, Switzerland). GL261 cells were injected intracranially (i. c.) in the right hemisphere using a stereotactic robot (Neurostar, Tubingen, Germany). Briefly, a blunt-ended syringe (Hamilton; 75N, 26s/2”/2, 5 µL) was placed 1.5 mm lateral and 1 mm frontal of bregma. The needle was lowered into the burr hole to a depth of 4 mm below the dura surface and retracted 1 mm to form a small reservoir. The injection was performed in a volume of 2 µL at 1 µL/min. A total of 20,000 cells in 2 μL PBS were injected at a speed of 1 μL/min intracranially into mice using a neurostar stereotaxic robot as described in Beffinger et al. 2019 [27]. After leaving the needle in place for 2 min, it was retracted at 1 mm/min. The burr hole was closed with bone wax (Aesculap, Braun, Tuttlingen, Germany) and the scalp wound was sealed with tissue glue (Indermil, Henkel, Düsseldorf, Germany). Anesthesia was interrupted using a mixture of flumazenil (Labatec Pharma AG, Geneva, Switzerland) and Buprenorphine (Indivior Schweiz AG, Zug, Switzerland), followed by injection of atipamezole 20 min later (Janssen). Carprofen (Pfizer AG, Zurich, Switzerland) was used for perioperative analgesia. Mice were collected for experiments at 20 days post glioma inoculation and the tumor-bearing hemispheres were processed with either mechanical dissociation at 4 °C or with enzymatic digestion at 37 °C to test the respective technique’s efficiency in producing single-cell suspensions for flow cytometric analysis (see below).

### 4.4. Brain Dissociation and Cell Isolation

The protocol has been optimized from our previous protocol [1] and the procedures provided by Miltenyi (adult brain dissociation kit, ABDK, Miltenyi, Bergisch Gladbach, Germany). The protocol is optimized for the dissociation and isolation of cells from brain tissue of small volume (e.g., mouse hippocampi). A detailed dissociation protocol for larger pools of tissue or whole adult mouse brain is provided in Appendix A. A detailed list of materials and reagents recommended for the following cell isolation protocol is displayed in Table 1 below.

The animals were deeply anesthetized with an overdose of Nembutal (Abbott Laboratories, North Chicago, IL, USA) and transcardially perfused with 15 mL ice-cold, calcium- and magnesium-free Dulbecco’s phosphate-buffered saline (DPBS, pH 7.3–7.4) via a 20 mL syringe and a 23 G needle (25 mm length). The brains were quickly removed and washed with ice-cold DPBS, after which the hippocampi were dissected on a cooled petri dish and placed in an ice-cold Hibernate-A medium.

For enzymatic digestion (ED) at 37 °C, 2–4 hippocampi/samples were grossly minced with surgical scissors and placed in a sterile 12-well plate containing dissection medium pre-heated to 37 °C. The dissection medium contained Roswell Park Memorial Institute (RPMI) medium, 10% fetal bovine serum (FBS), 0.4mg/mL collagenase and 2 mg/mL DNaseI. The plate was then placed in a standard cell culture incubator at 37 °C with 95% O_2_ and 5% CO_2_ for a total of 30 min. After the first 15 min of incubation, the tissue was mixed with a pipette to optimize the enzymatic digestion. At the end of the 30 min enzymatic digestion, the plate was placed on ice, and the digested tissue was transferred to a 1 mL Dounce homogenizer to complete the dissociation of remaining tissue pieces (on ice). Thereafter, the homogenate was sieved through a 70 μm cell strainer mounted onto a 50 mL Falcon tube and transferred into 5 mL Eppendorf tubes on ice. Eppendorf tubes made of polypropylene were used because cells show less adherence to this material as opposed to polystyrene [5].

Mechanical dissociation (MD) at 4 °C was carried out on ice, while all the solutions were kept at 4 °C. 2–4 hippocampi/sample were dissociated in 1.5 mL Hibernate-A medium in a 1 mL Dounce homogenizer with a loose pestle. The 1 mL Dounce homogenizer used here (see Table 1) has enough capacity for a volume of 1.5 mL, which is preferable for optimal tissue dissociation. Furthermore, this mechanical homogenizer allows enough space between the glass pot and the loose pestle, which is required for efficient tissue homogenization without extensive cell loss. The tissue was gently homogenized until no bigger tissue pieces were visible. The homogenized tissue was then sieved through a 70 μm cell strainer mounted onto a 50 mL Falcon tube. The Dounce homogenizer was then washed twice with 1 mL Hibernate-A, whereby each wash is poured onto the cell strainer. The homogenized hippocampi were then transferred to 5 mL Eppendorf tubes and kept on ice as described above.

After ED at 37 °C or MD at 4 °C, all samples were handled the same way and further processed as follows: The homogenates were pelleted at 400× *g* for 6 min at 4 °C in a swing-bucket rotor centrifuge (Eppendorf, Schönenbuch, Switzerland). The supernatants were removed and 1 mL ice-cold DPBS (pH 7.3–7.4) was added to all samples. The pellets were then re-suspended with a P1000 micropipette, applying a pipette-tip cut-off. After re-suspension, the final volume in each tube was brought to 1.5 mL. 500 μL of freshly prepared isotonic percoll solution (percoll, GE Healthcare, Opfikon, Switzerland) was then added to each sample (final volume: 2 mL) and mixed well (applying a pipette-tip cut-off to optimize the mixing). Percoll was rendered isotonic by mixing 1 part of 10 × calcium- and magnesium-free DPBS (pH 7.3–7.4) with 9-parts of percoll. Importantly, the pH of percoll was adjusted to 7.3–7.4 with 5 molar hydrochloric acid before starting the isolation procedure. The percoll solution was mixed properly with the cell suspension, after which 2 mL of DPBS were gently layered on top of it with a pipette boy set on the slowest speed, creating two separate layers. The samples were centrifuged for 10 min at 3000× *g*. The centrifugation resulted in an upper layer consisting of DPBS and a lower layer consisting of percoll. The two layers were separated by a disk of myelin and debris, while the cells were located at the bottom of the tube. The layers were aspirated, leaving about 500 μL as some cells, depending on their size, usually float in percoll just above the pellet. The cells were then washed once in DPBS making sure not to resuspend the pellet. This was achieved by gently adding 4 mL DPBS, closing the tube and holding it in a horizontal position, and gently tilting it 145 degrees to mix the remaining percoll with the added DPBS. The cells were then pelleted by centrifuging them at 400× *g* for 10 min at 4 °C.

In Appendix A, we show the utility and effectiveness of the chosen brain dissociation and cell isolation protocol in obtaining a large amount of microglia and astrocytes from as little as 1 single hippocampus from 1 brain hemisphere of adult male mice. Indeed, enough RNA can be obtained from either microglia or astrocytes from 1 hippocampus to perform scSeq, or to generate an RNA library for bulk sequencing (Appendix A).

### 4.5. Microglia and Astrocyte Isolation

A schematic illustration of the procedures used for microglia and astrocyte isolation is provided in Figure 3a. For the proteomic analysis, microglia or astrocytes were isolated from 4 hippocampi (2 mice/sample) via magnetic-activated cell sorting (MACS) using mouse anti-CD11b (for microglia) or anti-ACSA-2 (for astrocytes) magnetic microbeads (Miltenyi, Bergisch Gladbach, Germany, see Table 1) according to the manufacturer’s instructions with some modifications. The MACS buffer used consisted of 1.5% bovine serum albumin (BSA) diluted in DPBS from a commercial 7.5% cell-culture grade BSA stock (Thermo Fisher Scientific). For the isolation of astrocytes, total hippocampal cell pellets after percoll (see above) were re-suspended in 80 μL MACS buffer and 10 μL FcR-blocking reagent (Miltenyi). The cells were then incubated for 10 min at 4 °C. Thereafter, 10 μL of anti-ACSA-2 microbeads were added and the cells were incubated for 15 min at 4 °C. The cells were then washed with 1 mL MACS buffer and pelleted at 300× *g* for 5 min at 4 °C. The cells were then passed through an MS MACS column attached to a magnet. This led ACSA-2-labeled cells to stay attached to the column, whereas unlabeled cells flowed through the column. After washing the columns three times with MACS buffer, astrocytes were flushed from the column with 1 mL MACS buffer and pelleted at 300× *g* for 5 min at 4 °C. Cell pellets were then snap-frozen in liquid nitrogen and stored at –80 °C. The same procedures (including the FcR-blocking step) were used to isolate microglia via anti-mouse CD11b microbeads. Appendix A show a qRT-PCR-based ascertainment of microglial or astrocytic cell enrichment after MACS from total brain cells obtained with the 4 °C protocol described above (See Appendix A for the detailed cell isolation protocol). Cells were counted manually using a standard hemocytometer (NanoEnTek, Seoul, South Korea, product code DHC-N01).

### 4.6. RNA Extraction and Quantification

The RNA was extracted via the Lexogen Split-RNA extraction kit (Lexogen, Vienna, Austria) according to the manufacturer’s instructions. The kit is based on phenol-chloroform extraction in acidic conditions, and we found this technique to yield the highest amount of RNA. RNA concentrations were measured via Qubit 4 fluorometer (Invitrogen, Carlsbad, CA, USA), using the RNA HS Assay kit (Invitrogen).

### 4.7. Quantitative Real-Time PCR

RNA was analyzed by TaqMan qRT-PCR instrument (CFX384 real-time system, Bio-Rad Laboratories, Cressier, Switzerland) using the iTaq™ Universal Probes One-Step Kit for probes (Bio-Rad Laboratories). The samples were run in 384-well formats in triplicates as multiplexed reactions with a normalizing internal control. We chose 36B4 as an internal standard for gene expression analyses. Thermal cycling was initiated with incubation at 50 °C for 10 min (RNA retrotranscription) and then at 95 °C for 5 min (TaqMan polymerase activation). After this initial step, 39 cycles of PCR were performed. Each PCR cycle consisted of heating the samples at 95 °C for 10 s to enable the melting process and then for 30 s at 60 °C for the annealing and extension reaction. Cutsom-made primers with probes for TaqMan were purchased from Thermo Fisher: housekeeping gene: 36B4, product code: NM_007475.5, Siglech, product code: Mm_00618627_m1, P2ry12, product code: Mm00446026_m1, Gfap, product code: Mm01253033_m1, Slc1a3, product code: Mm00600697_m1. The relative target gene expression was calculated according to the Delta C(T) method.

### 4.8. Single-cell RNA Sequencing (scRNA-seq) using 10X Genomics Platform

The quality and concentration of the single-cell preparations were evaluated using a hemocytometer in a Leica DM IL LED microscope and adjusted to 1000 cells/µL. 10,000 cells per sample were loaded into the 10X Chromium controller (10X Genomics, Pleasanton, CA, USA) and library preparation was performed according to the manufacturer’s indications (single-cell 3′ v3 protocol). The resulting libraries were sequenced in an Illumina NovaSeq sequencer according to 10X Genomics recommendations (paired-end reads, R1 = 28, i7 = 8, R2 = 91) to a depth of around 50,000 reads per cell. The quality check outcomes are displayed in Appendix A.

### 4.9. Single-cell RNA-Seq Analysis

10 x Chromium data were demultiplexed with cellranger mkfastq v2.0.2. Gene-cell count matrix was generated by cellranger count v2.0.2 against Ensembl GRCm38.p5 reference genome.

Single Cell RNA-Seq data analysis was carried out with R Seurat (V3) package. Specifically, to account for potential batch effects, we used canonical correlation analysis (utilized in IntegrateData function) that identifies a linear combination of features to construct a shared correlation structure and align the global transcriptome across 2 datasets [28,29]. Only genes that were detected in at least 5 cells were included in the analysis. Furthermore, we excluded from the analysis of all cells with less than 200 or more than 5000 genes detected, and cells that have more than 25% mitochondrial fraction. We carried out standard preprocessing (log-normalization) and identified the top 2000 variable features for each dataset (ED37 °C and MD4 °C). We then identified integration anchors using the FindIntegrationAnchors function. Here we used all default parameters for identifying anchors between the two data sets, setting the ‘dimensionality’ to 1:15. The final batch-corrected expression matrix was created from this anchor-set using IntegrateData function where the number of PCs used for weighting was set to 15. We employed a standard Seurat workflow for clustering and visualization: data scaling, PCA analysis, and UMAP clustering using PCA reduction. For improved visualization, we further filtered out all cells located more than 3 standard deviations away from their cluster center and those which had a different identity than the majority of their 10 nearest neighbors in the UMAP.

The differential expression analysis was carried out using the FindMarker function (logfc.threshold = 0). *p*-values were calculated using the Wilcoxon Rank Sum test and adjusted with Bonferroni correction

### 4.10. Code Availability

A detailed list of commands used to analyze the single-cell sequencing data as well as gene count matrices are available at https://github.com/bihealth/SC-RNA-Seq-37vs4.

### 4.11. Proteotype Analysis

#### 4.11.1. Liquid Chromatography–Tandem Mass Spectrometry (LC–MS/MS) Analysis

The samples used for proteotype analysis were prepared using S-trap (Protifi, Huntington, NY, USA) columns according to the manufacturer’s instructions. For MS analysis, peptides were reconstituted in 5% acetonitrile and 0.1% formic acid containing iRT peptides (Biognosys AG, Schlieren, Switzerland) as described in Escher et al., 2012 [30].

The peptides resulting from the isolation of microglia were analyzed in data-independent acquisition (DIA) and data-dependent acquisition (DDA) mode for spectral library generation. For spectral library generation, a fraction of the samples originating from the same condition were pooled to generate mixed pools for each condition. Peptides were separated by reverse-phase chromatography on a high-pressure liquid chromatography (HPLC) column (75-μm inner diameter; New Objective, Littleton, MA, USA) packed in-house with a 50-cm stationary phase ReproSil-Pur 120A C18 1.9 µm (Dr. Maisch GmbH, Ammerbuch, Germany) and connected to an EASY-nLC 1000 instrument equipped with an autosampler (Thermo Fisher Scientific). The HPLC was coupled to a Fusion mass spectrometer equipped with a nanoelectrospray ion source (Thermo Fisher Scientific). Peptides were loaded onto the column with 100% buffer A (99% H2O, 0.1% formic acid) and eluted with increasing buffer B (99.9% acetonitrile, 0.1% formic acid) over a nonlinear gradient for 120 min. The DIA method (Bruderer et al. 2017) [31] contained 26 DIA segments of 30,000 resolution with IT set to 60ms, AGC of 3 × 10^6^, and a survey scan of 120,000 resolution with 60 ms max IT and AGC of 3 × 10^6^. The mass range was set to 350–1650 m/z. The default charge state was set to 2. Loop count 1 and the normalized collision energy was stepped at 27. For the DDA, a 3 s cycle time method was recorded with 120,000 resolution of the MS1 scan and 20 ms max IT and AGC of 1 × 10^6^. The MS2 scan was recorded with 15,000 resolution of the MS1 scan and 120 ms max IT and AGC of 5 × 10^4^. The covered mass range was identical to the DIA.

The peptides resulting from the isolation of astrocytes were analyzed in the DIA and DDA mode for spectral library generation. For spectral library generation, a fraction of the samples originating from the same condition were pooled to generate mixed pools for each condition. Peptides were separated by reverse-phase chromatography on a 50 cm EASY-Spray C18 LC column (Thermo Fisher Scientific) connected to an EASY-nLC 1200 instrument equipped with an autosampler (Thermo Fisher Scientific). The HPLC was coupled to a Fusion Lumos mass spectrometer equipped with a nanoelectrospray ion source (Thermo Fisher Scientific). Peptides were loaded onto the column with 100% buffer A (99% H_2_O, 0.1% formic acid) and eluted with increasing buffer B (80% acetonitrile, 0.1% formic acid) over a nonlinear gradient for 120 min. The DIA method [32] contained 40 DIA segments of 30,000 resolution with IT set to 55 ms, AGC of 1 × 10^6^, and a survey scan of 120,000 resolution with 50 ms max IT and AGC of 5 × 10^5^. The mass range was set to 350–1650 m/z. The default charge state was set to 2. Loop count 1 and the normalized collision energy was set to 27. For the DDA, a 3 s cycle time method was recorded with 120,000 resolution of the MS1 scan and 25 ms max IT and AGC of 5 × 10^5^. The MS2 scan was recorded with 15,000 resolution of the MS1 scan and 35 ms max IT and AGC of 2 × 10^4^. The covered mass range was identical to the DIA.

#### 4.11.2. Data Analysis DIA LC-MS/MS

LC-MS/MS DIA runs were analyzed with Spectronaut Pulsar X version 12 (Biognosys) [31] using default settings. Briefly, a spectral library was generated from pooled samples measured in DDA (details above). The collected DDA spectra were searched against UniprotKB (UniProt Swiss-prot, Mus musculus retrieved 2018) using the Sequest HT search engine within Thermo Proteome Discoverer version 2.1 (Thermo Fisher Scientific). We allowed up to two missed cleavages and semi-specific tryptic digestion. Carbamidomethylation was set as fixed modification for cysteine, oxidation of methionine, and deamidation of arginine were set as variable modifications. Monoisotopic peptide tolerance was set to 10 ppm, and fragment mass tolerance was set to 0.02 Da. The identified proteins were assessed using Percolator and filtered using the high peptide confidence setting in Protein Discoverer. Analysis results were then imported to Spectronaut Pulsar version 12 (Biognosys AG, Schlieren, Switzerland) for the generation of spectral libraries.

Targeted data extraction of DIA-MS acquisitions was performed with Spectronaut version 12 (Biognosys AG) with default settings using the generated spectral libraries as previously described [31]. The proteotypicity filter “only protein group-specific” was applied. Extracted features were exported from Spectronaut for statistical analysis with MSstats (version 3.8.6) using default settings [33]. Briefly, features were filtered for calculation of Protein Group Quantity as defined in Spectronaut settings, common contaminants were excluded. For each protein, features were log-transformed and fitted to a mixed effect linear regression model for each sample in MSstats [33]. In MSstats, the model estimated fold change and statistical significance for all compared conditions. Significantly different proteins were determined by the threshold fold-change > 2 and adjusted *p*-value < 0.01. Benjamini–Hochberg method was used to account for multiple testing.

### 4.12. Gene Ontology Analysis

The functional enrichment analysis for the transcriptome data was carried out with R tmod package [34], using Utest. For each cell type, we used genes that were detected by Seurat::FindMarker (logfc.threshold = 0) function and sorted them by adjusted *p*-values: microglia—5631 genes, astrocytes—4108 genes, and neurons—3288 genes. The enrichment analysis is done using the GO gene set collection from MsigDB. To match MsigDB gene symbols, mouse gene ids were converted to upper case. Significantly different genes were determined by the threshold: log-fold change > 0.5, adjusted *p*-value cut-off < 0.01. Appendix A contains a full list of deregulated genes for all cells and Appendix A contains the full GO-analysis of deregulated genes for each cell type analyzed. For the proteomic GO analysis, we used a hypergeometric test from the GOrilla online tool [35]. For the proteomic GO analysis, we used the hypergeometric test from GOrilla online tool: all quantified microglial or astrocytic proteins were used as background, while the significantly deregulated proteins were used as target set (all as Uniprot IDs, log-fold change > 2, adjusted *p*-value cut-off < 0.01). Significantly up- and downregulated proteins were analyzed separately using the same background set of proteins. Appendix A contains the full list of deregulated proteins and GO analysis per cell type, respectively.

### 4.13. Comparison of RNA and Protein Expression Changes

In astrocytes, we quantified 3823 proteins present in both conditions (ED37° and MD4°), which were assigned to 3623 unique genes. Median log-fold changes (LFCs) were used for the comparison of RNA and protein expression changes. For microglia, we quantified 4328 proteins that were present in both conditions (ED37° and MD4°), and these were assigned to 4158 unique genes. Pearson’s correlation was used to assess the linear association between the datasets.

### 4.14. Flow Cytometric Analysis

Data analysis was performed using FlowJo 10.5.3 (BD, Franklin Lakes, NJ, USA). Populations of interest were manually pre-gated in FlowJo software. Microglia were identified as CD45^medium^ CD11b^medium^, live, single cells.

After preprocessing we combined an equal number of randomly selected cells from each group and visualized data using t-Distributed Stochastic Neighbor Embedding (t-SNE). The reagents used in flow cytometry are summarized in Table 2 below.

One hippocampus/sample (from 1 brain hemisphere) was used for the flow cytometric experiments. Cells were isolated as described above. After cell isolation, the samples were washed with PBS and stained for surface markers and live/dead reagent. Following washing with PBS, the cells were fixed and permeabilized with Cytofix/Cytoperm (BD, #554715), followed by staining for intracellular markers. Flow cytometry was performed on an LSR II Fortessa (equipped with 405 nm, 488 nm, 561 nm, and 640 nm laser lines; special order research product, BD) with FACS Diva Software. Before the acquisition, PMT voltages were manually adjusted to reduce fluorescence spillover, and single-stain controls were acquired for compensation matrix calculation. The specific antibodies used for the flow cytometric staining are listed in Table 2 below.

### 4.15. List of Abbreviations

ED37°, enzymatic digestion at 37 °C; MD4°, mechanical dissociation at 4 °C; RT, room temperature; scRNA-Seq, single-cell RNA-sequencing; UMAP, uniform manifold approximation and projection; GO-analysis, gene ontology analysis; FACS, fluorescence-activated cell sorting; DIA, data-independent acquisition; DDA, data-dependent acquisition; LC-MS/MS, liquid chromatography–tandem mass spectrometry; PE, phycoerythrin; APC, allophycocyanin; BV, brilliant violet; PerCP, Peridinin-Chlorophyll-Protein; CD, cluster of differentiation; SIRPα, Signal regulatory protein α; FcγR, Fc-gamma receptor; tSNE, t-distributed stochastic neighbor embedding; ICS, intracellular staining; PBS, phosphate buffered saline; PMT, photomultiplier tubes; MHC, major histocompatibility complex.

### 4.16. Accessibility and Availability of Data and Materials

The raw data for the single-cell sequencing have been deposited in NCBI’s Gene Expression Omnibus (Edgar et al., 2002, [36]) and are accessible through GEO Series accession number GSE143796 (https://www.ncbi.nlm.nih.gov/geo/query/acc.cgi?acc=GSE143796).

All mass spectrometric data and acquisition information were deposited to the ProteomeXchange Consortium (www.proteomexchange.org/) via the PRIDE partner repository [37] (data set identifier: PXD015592, username: reviewer08386@ebi.ac.uk, password: kRa2alY7).

## Figures and Tables

**Figure 1 ijms-21-07944-f001:**
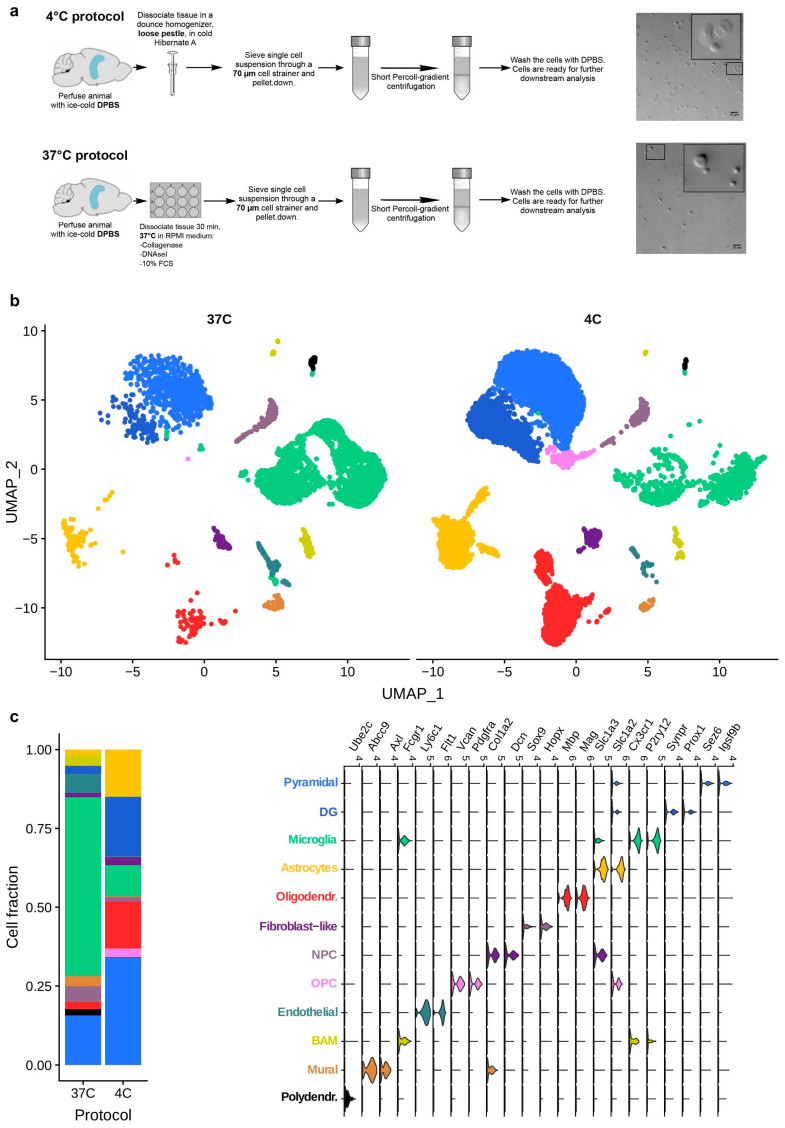
The cell isolation method affects the proportions of murine hippocampal cell populations. (**a**) Schematic illustration of the two cell isolation methods. Adult murine hippocampi were mechanically dissociated at 4 °C (MD4°) or digested enzymatically for 30-min at 37 °C (ED37°). After myelin removal, total hippocampal cells from both conditions were subjected to scRNA-seq. The photomicrographs show the shape of cells isolated via MD4° or ED37°. Note that cells isolated via MD4° appear larger and display discernible nuclei and cytoplasm (Scale bar: 25 µm). (**b**) Uniform manifold approximation and projection (UMAP) scores showing the clustering of cells isolated via ED37° (4448 cells) or MD4° (11868 cells). (**c**) Colored legends (**left**) represent the identities and total fractions for each cell type and the violin plots (**right**) show the cell identity based on the enriched expression of specific genes.

**Figure 2 ijms-21-07944-f002:**
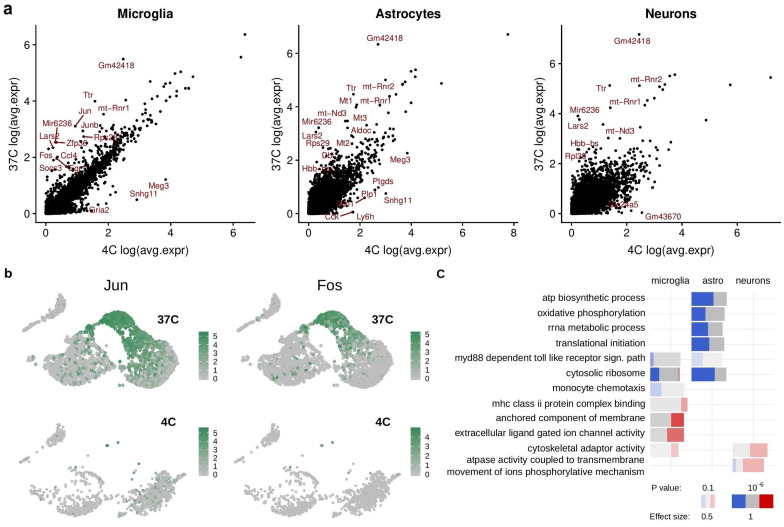
The cell isolation protocol alters the transcriptional profile of mouse hippocampal glia and neuronal cells. (**a**) Scatter plots of the differential gene expression in microglia, astrocytes, and neurons obtained via ED37° relative to MD4°. Some examples of up- and downregulated genes following ED are highlighted in red in the scatterplots. (**b**) Feature plot magnification of the microglial population from adult mouse hippocampi displaying the expression of the immediate-early genes *Jun* and *Fos*. ED37 °C induces the appearance of a microglial population characterized by increased expression of immediate early genes (upper panel) which is barely present in microglia from hippocampi processed via MD4° (lower panel). (**c**) Selected gene ontology (GO) terms associated with significantly deregulated genes in microglia, astrocytes, and neurons (ED37° relative to MD4°). The bar color represents downregulation in ED37° relative to MD4° (blue) and upregulation in ED37° relative to MD4° (red). The intensity of the respective color indicates the adjusted *p*-value, while the size of the bars denotes the effect size, i.e., the area under the curve (AUC, see Appendix A). Significantly different genes were determined by the threshold: log-fold change > 0.5, adjusted *p*-value cut-off < 0.01.

**Figure 3 ijms-21-07944-f003:**
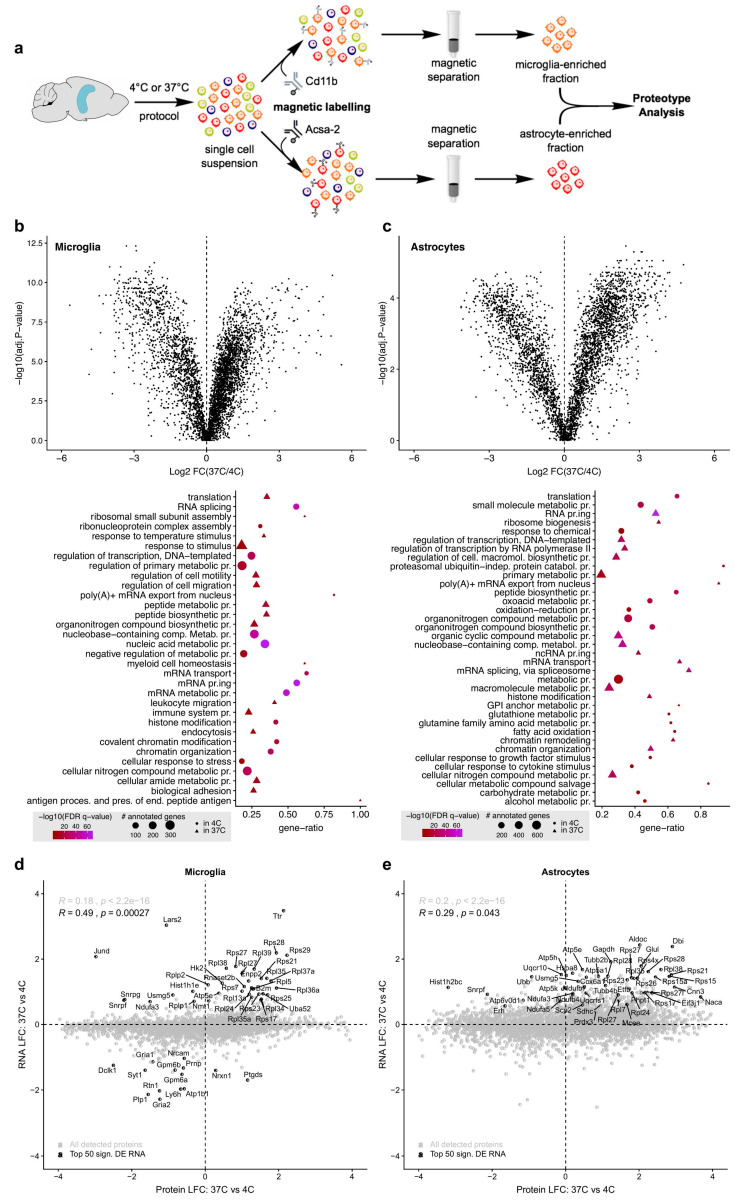
The cell isolation method affects the proteotype profiles of mouse microglia and astrocytes. (**a**) Schematic illustration of the main experimental procedure. Adult murine hippocampi were mechanically dissociated at 4 °C (MD4°) or digested enzymatically for 30 min at 37 °C (ED37°). After myelin removal, microglia or astrocytes where freshly isolated via magnetic associated cell sorting (MACS) for cell-specific proteotype analysis. (**b**,**c**) Volcano plots (**top**) and select gene ontology (GO) terms (**bottom**) of deregulated proteins in (**b**) microglia and (**c**) astrocytes obtained via ED37° relative to MD4°. *N* = 4 biological replicates/group. Significantly different proteins were determined by the threshold: fold-change > 2 and adjusted *p*-value < 0.01. Benjamini–Hochberg method was used to account for multiple testing. (**d**,**e**) Comparison of (**d**) microglial and (**e**) astrocytic RNA and protein expression differences between the two protocols. The top 50 differentially expressed genes (sorted by RNA adj. *p*-value), which were detected at both RNA and protein levels, are highlighted in the scatterplots. A significant correlation was detected for both microglia (R = 0.49, *p* = 0.00027) and astrocytes (R = 0.29, *p* = 0.043).

**Figure 4 ijms-21-07944-f004:**
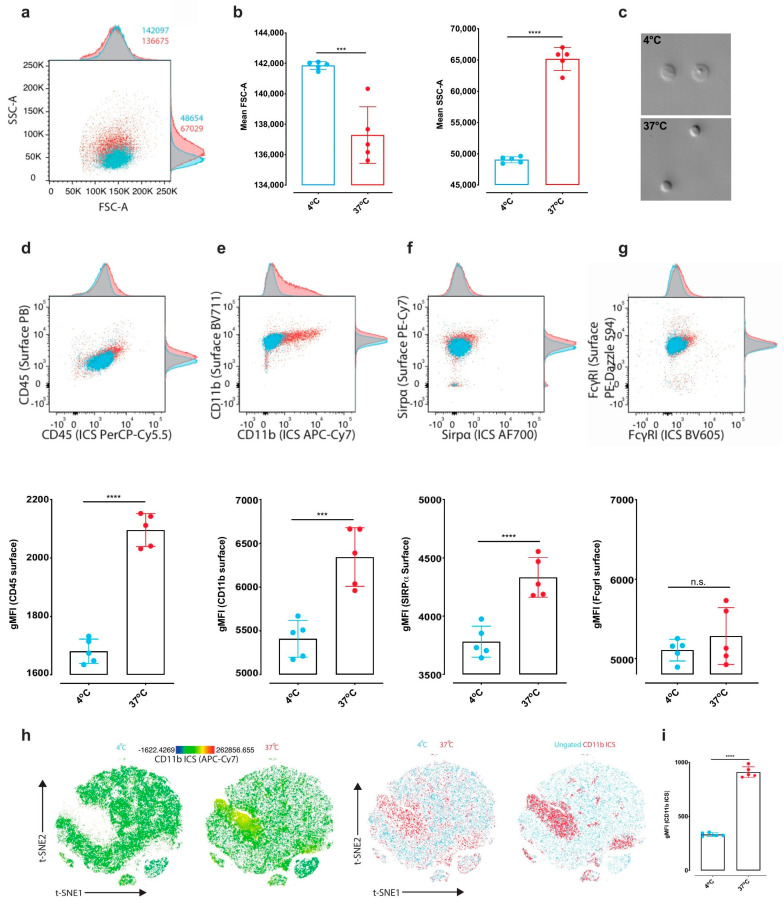
Cell isolation method affects microglial surface and intracellular marker expression in flow cytometric analysis. Flow cytometric analysis of surface and intracellular marker expression levels on microglia after ED37° versus MD4°. (**a**) Representative 2D-flow cytometric plot overlay with adjunct histograms and (**b**) scatterplots comparing forward scatter (FSC-A, t_8_ = 5.439, *p* = 0.0006) and side scatter (SSC-A, t_8_ = 18, *p* < 0.0001). (**c**) Photomicrographs illustrating the cell shape after MD4° or ED4° (20x magnification). (**d**–**g**) Representative 2D-flow cytometric plot overlays with adjunct histograms and scatter plots comparing geometric mean fluorescence intensities (gMFIs) of surface and intracellular expression of (**d**) CD45 (t_8_ = 13.28, *p* < 0.0001), (**e**) CD11b (t_8_ = 5.288, *p* = 0.0007), (**f**) SIRPα (t_8_ = 5.213, *p* = 0.0004) and (**g**) FcγRI, (t_8_ = 1.030, *p* = 0.3). (**h**) A t-SNE clustering algorithm was used to depict CD11b expression in different populations of microglia. Left row showing relative intracellular CD11b expression levels for both conditions. The right row shows an overlay of the two conditions, as well as all populations overlaid with high CD11b ICS expression. (**i**) Scatterplot for the quantification of the CD11b intracellular signal for microglia isolated via MD4° versus ED37° (t_8_ = 24.43, *p* < 0.0001) Summary of two independent experiments, *n* = 4 and 5 biological replicates/group respectively. Unpaired two-tailed Student *t*-tests were used to compare the means. *** *p* < 0.001, **** *p* < 0.0001. Error bars represent the mean ± standard deviation.

**Table 1 ijms-21-07944-t001:** List of material and reagents for the cell isolation procedure.

Material	Product Code
20 mL Syringes, Bbraun	4606205V
23 G needles, 25 mm Sterican	4657667
Petri dishes, Thermo Fisher Scientific	11339283
5 mL Tubes, Eppendorf	0030119452
1x DPBS, Thermo Fisher Scientific	14190144
10x DPBS, Thermo Fisher Scientific	14200075
Percoll, GE Healthcare	17089101
1 mL Dounce Homogenizer, Active Motif	40401
Hibernate A, Brainbits	HAPR
70 μm cell strainers, Fisher Scientific	15370801
Collagenase, Sigma Aldrich/Merck	10269638001
DNase I, Sigma Aldrich/Merck	10104159001
RPMI medium, Sigma Life Science	R8758–500ml
MACS MS Columns, Miltenyi	130-042-201
CD11b magnetic beads, Miltenyi	130-093-634
ACSA-II magnetic beads, Miltenyi	130-097-678
7.5% BSA in PBS, Thermo Fisher Scientific	11500496
FBS, Thermo Fisher Scientific	26140087
Swing Bucket Centrifuge 5810, Eppendorf	5810000010

**Table 2 ijms-21-07944-t002:** List of antibodies used for the flow cytometric analysis.

Antigen	Clone	Color	Vendor	Product #
CD11b	M1/70	BV711	Biolegend	101242
CD11b	M1/70	APC-Cy7	Biolegend	101226
CD45	30-F11	Pacific Blue	Biolegend	103126
CD45	30-F11	PerCP/Cy5.5	Biolegend	103131
FcγRI (CD64)	X54-5/7.1	BV605	Biolegend	139323
FcγRI (CD64)	X54-5/7.1	PE-Dazzle594	Biolegend	139319
CD172a (SIRPα)	P84	PE-Cy7	Biolegend	144007
CD172a (SIRPα)	P84	Alexa Fluor 700	Biolegend	144021
Fixable Zombie Aqua Viability Kit	N/A	AmCyan	Biolegend	423102

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
