# Peer review of "Enzymatic Dissociation Induces Transcriptional and Proteotype Bias in Brain Cell Populations"

_ijms, 2020, doi:10.3390/ijms21217944_

Round 1

Reviewer 1 Report

In this paper the Authors describe the influence of two main brain cell isolation protocols from adult mice on transcriptional and proteotype profiles of the different cell populations (glia, neurons, endothelial cells, fibroblasts, macrophages), analyzing single-cell transcriptomics, proteotypes of microglia and astrocytes and FACS analysis of microglia. The work is well performed and very interesting, aiding in revising published literature as well as in setting up conditions and subsequent data interpretation of transcriptomic and proteomic profiles of freshly isolated brain cells.

It would be worth considering several minor issues:

1) The purpose and rationale of analyzing the dissociation of brain tissue injected with glioma, should be introduced in the first paragraph (Introduction)

2) Lines from 317 to 328 are pasted instructions and not methods!

3) It would be interesting to better appreciate the different cell morphology obtained by the two methods, as it is quite difficult with the image proposed in Fig. 1a. In my opinion, it would be worth replcing the low-magnification photomicrographs with the high-magnification ones, possibly including a greater number of cells than the present ones.

4) Please add in the schematic protocol represented in Fig. 1a that in ED, after digestion, the tissues are also subjected to Dounce homogenization

5) Maybe a further discussion of which gene categories are almost up-regulated and which are down-regulated particularly in astrocytes and microglia will render this paper even more informative.

Author Response

REVIEWER #1

In this paper the Authors describe the influence of two main brain cell isolation protocols from adult mice on transcriptional and proteotype profiles of the different cell populations (glia, neurons, endothelial cells, fibroblasts, macrophages), analyzing single-cell transcriptomics, proteotypes of microglia and astrocytes and FACS analysis of microglia. The work is well performed and very interesting, aiding in revising published literature as well as in setting up conditions and subsequent data interpretation of transcriptomic and proteomic profiles of freshly isolated brain cells.

It would be worth considering several minor issues:

  • The purpose and rationale of analyzing the dissociation of brain tissue injected with glioma, should be introduced in the first paragraph (Introduction).

We thank the reviewer for pointing this out. We have added a sentence, lines 87-90, to introduce the application of our methods to glioma specimens. We have also included in the introduction a sentence regarding the general relevance of testing the effects of enzymatic dissociation at 37°C on FACS-based analyses (lines 64-68). Finally, we have included a sentence regarding the efficiency of ED and MD in producing single cell suspensions from brain tumor tissue in the result section, specifically for microglia cells (lines 192-198).

  • Lines from 317 to 328 are pasted instructions and not methods!

We thank the reviewer for bringing this to our attention, and we apologize for the confusion! The instructions have been removed.

  • It would be interesting to better appreciate the different cell morphology obtained by the two methods, as it is quite difficult with the image proposed in Fig. 1a. In my opinion, it would be worth replcing the low-magnification photomicrographs with the high-magnification ones, possibly including a greater number of cells than the present ones.

Although reviewer 1 raises a fair point, we would like to keep the representative images in their present form, as the low magnification pictures also provide a representative qualitative overview of the higher cellular density obtained through the MD protocol as compared to the ED procedure, which is also reflected in the outcome of the scSeq.

  • Please add in the schematic protocol represented in Fig. 1a that in ED, after digestion, the tissues are also subjected to Dounce homogenization.

We modified Fig. 1a as suggested by the reviewer, adding the final round of Dounce homogenization following the enzymatic digestion.

  • Maybe a further discussion of which gene categories are almost up-regulated and which are down-regulated particularly in astrocytes and microglia will render this paper even more informative.

We complemented the discussion with respect to microglial and astrocytic alterations as suggested by reviewer 1 with a new paragraph (lines 205-226).

Reviewer 2 Report

Currently, I want to congratulate the authors for the effort to publish this set of exciting data. Sometimes, negative and the evolution of data is extremely important. Your manuscript is a perfect example. 

There are few mistakes that need correction and check the past or present tense

Thank you great work

Author Response

REF. ijms-956759

Manuscript Title: “Enzymatic dissociation induces transcriptional and proteotype bias in microglia and other brain cells” by Mattei et al.

Point-to-point rebuttal

REVIEWER #2 
Currently, I want to congratulate the authors for the effort to publish this set of exciting data. Sometimes, negative and the evolution of data is extremely important. Your manuscript is a perfect example. There are few mistakes that need correction and check the past or present tens. Thank you great work.

The authors would really like to thank the Reviewer for the kind and positive feedback.

Reviewer 3 Report

This work compared enzymatic and mechanical dissociation of brain tissue effects on gene expression of glial cells.  Neurons were not reported.  The analyses of cells resulting from the two methods is comprehensive and careful.  The major problem is that the two dissociation methods were carried out at different temperatures so it is not possible to come to any confident conclusion regarding the relative effects of mechanical and enzymatic dissociation.  The mechanical dissociation should have been under the same temperature conditions as the enzymatic dissociation.

Author Response

REF. ijms-956759

Manuscript Title: “Enzymatic dissociation induces transcriptional and proteotype bias in microglia and other brain cells” by Mattei et al.

Point-to-point rebuttal

REVIEWER #3

This work compared enzymatic and mechanical dissociation of brain tissue effects on gene expression of glial cells.  Neurons were not reported.  The analyses of cells resulting from the two methods is comprehensive and careful.  The major problem is that the two dissociation methods were carried out at different temperatures so it is not possible to come to any confident conclusion regarding the relative effects of mechanical and enzymatic dissociation.  The mechanical dissociation should have been under the same temperature conditions as the enzymatic dissociation.

We thank the Reviewer for his/her comments.

We agree that mechanical dissociation at 37°C would have been an interesting addition for the sake of comparison and that this condition would be useful in order to discern the technical effects of the dissociation per se, from the temperature component. However, since the standard enzymatic digestion is commonly performed at 37°C, we sought to compare this method -as a whole- with an alternative one based on mechanical dissociation, which can be instead performed at 4°C.

Thus, we agree with the Reviewer that in the present work, it is not possible to dissect the specific contribution of the temperature to the observed biological artifacts induced by the enzymatic-based method. The limitations of the experimental design have been now overtly discussed (lines 229-232), highlighting that with the current results it is not possible to dissect the specific cellular responses induced by the thermal component from those induced by the enzymatic digestion. Furthermore, to emphasize that the comparison is between two methods, and not only between temperature conditions- we now refer to the enzymatic digestion as to “ED37°” and to mechanical dissociation as to “MD4°” throughout the entire manuscript and in the supplementary information. Overall, the findings we report provide comprehensive evidence for technical biases and biological artifacts when implementing enzymatic digestion-based isolation methods for brain cell analyses.

Round 2

Reviewer 3 Report

Although the authors have made only minor changes in their manuscript, I think the changes they have made and the response to my comments make it clear that their work addresses a comparison of two relatively standard methods of dissociation, and does not address the components (e.g. temperature) of these methods. Therefore, I now find this manuscript acceptable for publication.